# Systematic Analysis of the Service Process and the Legislative and Regulatory Environment for a Pharmacist-Provided Naltrexone Injection Service in Wisconsin

**DOI:** 10.3390/pharmacy7020059

**Published:** 2019-06-12

**Authors:** James H. Ford, Aaron Gilson, David A. Mott

**Affiliations:** Sonderegger Research Center, University of Wisconsin School of Pharmacy, Madison, WI 53705, USA; jhfordii@wisc.edu (J.H.F.II); aaron.gilson@wisc.edu (A.G.)

**Keywords:** naltrexone, opioid use disorder, implementation, service process, regulatory, community pharmacy

## Abstract

Community pharmacists are viewed by the public as convenient and trustworthy sources of healthcare and pharmacists likely can play a larger role in addressing the major public health issue of the opioid epidemic affecting Wisconsin residents. Approved medications, including long-acting injectable naltrexone, can transform the treatment of individuals with opioid use disorder (OUD). Due to shortages of behavioral health providers in the U.S., and pharmacists’ knowledge about the safe use of medications, pharmacists can be a significant access point for treating OUD with naltrexone. Wisconsin’s pharmacy practice laws authorize pharmacists to administer medications via injection, and a small number of pharmacists currently are using this authority to provide a naltrexone injection service. This exploratory study had two objectives: (1) describe the pharmacist injection service process and identify barriers and facilitators to that service and (2) analyze the legislative/regulatory environment to ascertain support for expanding naltrexone injection service. Semi-structured pharmacist interviews (n = 4), and an analysis of Wisconsin statutes/regulations governing public health and social services, were undertaken to explore the objectives. Findings suggest that the service process requires considerable coordination and communication with practitioners, patients, and pharmacy staff, but many opportunities exist to broaden and sustain the service throughout Wisconsin.

## 1. Introduction

The current opioid epidemic, including prescription opioid misuse or abuse and associated overdose deaths, represents a major public health issue in the United States [1,2,3]. The impact of opioid-related harms, including overdoses, is significant. Nationally, the opioid epidemic has resulted in an increase in inpatient stays and emergency department visits [4,5,6,7,8,9]. The rate of opioid related deaths also increased by 345% from 2001 to 2016 with the percent of deaths attributable to opioids increasing from 0.4% to 1.5%, a 292% increase over the same time period [10]. The increase in deaths has led to a decline in life expectancy for the third consecutive year due in part to opioid use disorder (OUD) [11]. An OUD represents a chronic relapsing condition similar to diabetes or hypertension, where individuals will experience patterns of treatment and abstinence from opioid use followed by a resumption of opioid use and relapse [12,13,14].

Wisconsin has not evaded this public health crisis. From 2006 to 2016, the number of Wisconsinites with an OUD tripled from 5828 to 20,590 [15]. During the same time, the number of inpatient hospital discharges per 100,000 residents associated with OUD increased 93% as compared to a 308% increase in OUD-related emergency department discharges [16]. Mortality associated with OUD for Wisconsinites is also increasing. Over a 15-year period (2001 to 2016), opioid-related deaths have increased by 529%, an increase primarily influenced by prescription opioids deaths [17]. The rate of change in opioid-related deaths experienced in Wisconsin from 2001 to 2016 is 53% higher than the national average [17].

### 1.1. Medication for Opioid Use Disorder

Potential access to FDA-approved medications is transforming how treatment is delivered to individuals with OUD. Medication for OUD (MOUD) includes methadone (an opioid agonist), buprenorphine (a partial opioid agonist) in all forms (sublingual, injectable or implantable), or oral naltrexone (tablet) or extended-release naltrexone (an injectable opioid antagonist, also known as Vivitrol^®^). The medications can be used to manage symptoms of opioid withdrawal or maintenance (methadone or buprenorphine) or abstinence from opioids (naltrexone) [14]. Managing OUD with medications has been shown to be more effective than treatment as usual [12,18,19]. For example, patients who receive oral naltrexone experienced reduced time in inpatient substance abuse and mental health treatment [20] or reduced opioid use [21]. More importantly, individuals who received six naltrexone injections experienced improvements in employment, mental health and psychosocial functioning, and reduced opioid craving and drug use [22]. Improving MOUD access is broadly considered a crucial public health strategy in confronting the opioid epidemic [23,24,25,26].

Despite demonstrated effectiveness as a treatment for OUD and federal agencies’ calls for increased access, MOUD is underused in the United States [27,28]. Current delivery models for MOUD rely on patients accessing MOUD in primary care physician’s offices or through a local addiction treatment provider. However, problems exist with current delivery models for MOUD, such as there not being enough practitioners who are registered (i.e., DATA-waived) with the Drug Enforcement Administration (DEA) to prescribe buprenorphine for office based opioid addiction treatment, negative practitioner attitudes about becoming DATA-waived, concerns about diversion of methadone and buprenorphine, insufficient infrastructure in practitioner offices/clinics to provide injections while maintaining patient anonymity, and lack of knowledge about naltrexone as a viable treatment modality for OUD [29,30,31,32,33,34,35,36,37,38,39,40]. This final issue is especially problematic, given that naltrexone is not a controlled substance. As a result, unlike both methadone and buprenorphine, any licensed physician can prescribe naltrexone without requiring an additional registration. Thus, new approaches are needed to improve access to and underutilization of MOUD.

### 1.2. Pharmacist Involvement in Patient Access to MOUD

Approaches to the treatment of individuals with OUD have largely eschewed pharmacists’ involvement despite calls that specifically suggest that pharmacists be involved in such efforts [41,42,43]. Pharmacists can especially facilitate provision of MOUD because pharmacists commonly are one of the most accessible health care providers in communities [44,45], they provide a myriad of patient care services that contribute to public health [46,47], and patients are very accepting of the services provided by pharmacists [46,47,48]. In Wisconsin, an emerging pharmacist service is naltrexone injections for patients with OUD.

Pharmacists licensed in Wisconsin have legal authority to become an active partner in the care team for treating patients with OUD. The authority for pharmacists to provide patient care services, which can include naltrexone injections, is codified in Wisconsin law governing both medical and pharmacy practice (see Table 1), and represents a significant facilitating factor for pharmacists to be involved in OUD treatment. Beginning in March 2016 (2015 WI Act 290), pharmacists have had the legal authority to administer non-vaccine medications via injection provided they comply with training and reporting requirements. Additionally, Wisconsin’s medical practice laws provide statutory authority for physicians to use a collaborative practice agreement to delegate patient care services to a pharmacist. Interpretation of the laws suggest that activities allowed under collaborative practice agreements between physicians and pharmacists would include the management of patients with OUD, including administering drugs via injection and/or administering oral drugs.

As Table 1 suggests, state legislatures, agencies, and regulatory boards play an important role in facilitating access to care for patients with OUD by creating and passing statutes and interpreting statutory language in regulations that have the potential to affect OUD treatment. As an employer, a provider of social services, and a payer of health care services, state government can influence greatly whether pharmacists can have an accepted role in the treatment process for patients with OUD, as well as the extent to which pharmacists are expected to be involved in such treatment. As such, the broad analysis of state statutes and regulations related to requirements for public health and human services can identify the potential demand for a pharmacist-provided naltrexone injection service. Understanding where and how pharmacists potentially can fit into the treatment processes for patients with OUD established by states is an important step in expanding the injection service across the state and/or the U.S.

#### **Objectives** 

Pharmacists providing naltrexone injections to patients with OUD is a new service and a limited number of Wisconsin pharmacists currently are providing naltrexone injections to patients with OUD. The first objective of this exploratory study was to describe how pharmacists are currently providing the injection service, the processes used by pharmacists to provide the injection service, and barriers and facilitators for the injection service. Exploring the process and implementation barriers and facilitators will provide areas of future research and provide strategies to expand the service to other pharmacists.

The second objective of this exploratory study was to examine the legislative and regulatory environments in Wisconsin that can influence pharmacist involvement in treatment for patients with OUD. The review of both statutes and regulations governing a variety of healthcare practices and state social health or welfare agencies was meant to highlight a number of provisions that could potentially facilitate or impede the expansion of pharmacist-provided naltrexone treatment services for patients with OUD. Exploring provisions currently in state law will provide suggestions for ways to expand pharmacist service for naltrexone injections to treat OUD.

## 2. Materials and Methods

### 2.1. Objective 1—Service Process and Implementation

#### 2.1.1. Interview Guide

Since the naltrexone injection service has been emerging in Wisconsin since 2016, qualitative research methods, including interviews, were used to collect and analyze data about the process and implementation strategies developed and used by pharmacists providing the service. We constructed a semi-structured interview guide for the interviews. The interview guide was developed to obtain descriptive information from pharmacists about how the service was being provided. The interview guide included questions about topics such as how patients were referred to the pharmacy, the process developed by the pharmacist to provide the service, costs to the pharmacy for itemized inputs needed for the service process, system, prescriber, patient, and pharmacy barriers and facilitators to accessing and implementing the service, and a question related to any miscellaneous items related to the development, implementation, and experience with providing the service. Question probes were included to obtain more details about various aspects of the service.

#### 2.1.2. Sample

A snowball sampling technique was used to identify and sample pharmacists who were providing the injection service to interview. Initially, we obtained the names and contact information for two pharmacists providing the service from staff members at the Pharmacy Society of Wisconsin. When we contacted the initial pharmacists, we obtained the names of additional pharmacists providing the service by asking those pharmacists for names of other pharmacists that were providing naltrexone injections in their community.

#### 2.1.3. Data Collection

We conducted four interviews with a purposeful sample of five pharmacists. Four of the five pharmacists were currently providing naltrexone injections in community pharmacy environments and the other pharmacist was planning the development of the service within a health system. All interviews took place between April and June 2018. Four pharmacists were practicing in urban areas and one pharmacist was practicing in a rural area of Wisconsin. One interview included two pharmacists providing the service and was conducted via telephone. The other three interviews were conducted in person, two were conducted in a private room at the University of Wisconsin School of Pharmacy and one was conducted at the pharmacy site. In the interviews with pharmacists, which were conducted conversationally, we asked about the naltrexone injection process at the community pharmacy with additional questions intended to seek clarification or learn more about specific process steps (e.g., interaction with behavioral health provider or actual injection process). The interviews were not recorded; however, each author took extensive notes during the interviews. Since this was neither conceptualized nor viewed by the IRB as research, a systematic data collection procedure was not necessary to meet the purpose of a quality improvement investigation.

#### 2.1.4. Data Analysis

The notes from the interviews were reviewed to identify specific points and themes related to the naltrexone injection process at each community pharmacy. Authors analyzed the content of the interviews separately and then met to achieve consensus about their interpretation of identified content, and to discuss and negotiate the classification and naming of the processes and thematic domains related to the naltrexone injection process derived from the interview data.

### 2.2. Objective 2—Analysis of Wisconsin Statutes and Regulations

The purpose of the legislative and regulatory analyses was to identify requirements in state law that have the potential to either facilitate, allow, or impede pharmacist services to provide treatment to patients with OUD throughout Wisconsin. A synopsis of all identified policy requirements was created, as well as the implication of those requirements on pharmacists’ roles in countering the current shortage of treatment services for patients with OUD in local communities.

#### 2.2.1. Identification of Wisconsin Laws

All potentially relevant statutes and regulations were identified using Lexis^®^ Academic, an electronic legal database available to all faculty, staff, and students at the University of Wisconsin—Madison. Identification of applicable language in all statutes and regulations involved two phases: (1) a keyword search of electronic text, and (2) manual review of potentially-relevant sections or subsections of laws. Relevant legislative statutes, as well as related regulations, which serve as the “sample” for this study objective, are listed in Table 2 and demonstrate the state agencies that are most central to the clinical issue of drug abuse treatment. For professional practice regulation and licensing laws, this search was confined to physicians, advanced practice nurses, and pharmacists, who are authorized to prescribe, dispense, or administer medications for chronic diseases or conditions. In addition to the specific laws in Table 2, an effort was made to determine applicable provisions that relate to the ability of practitioners to engage in telehealth (i.e., using electronic communication for exchanging health information to facilitate patient care) around OUD-related issues.

#### 2.2.2. Policy Analysis

All content of the policies contained in Table 2 was collected, downloaded, and reviewed to select provisions that could be related to treatment with naltrexone. Regulation and Licensing laws, specifically pharmacy practice statutes and regulations, were examined first since they provide the legal authority for pharmacists’ involvement in treatment of patients with OUD (see Table 1). The other areas of legislation and regulations identified in Table 2 were then analyzed to determine the other areas in law that likely impact pharmacists’ roles in treatment of patients with OUD. Since state statutes provide the authority for state agencies to engage in certain activities, and regulations implement the statutory authority by defining the process for engaging in those activities, whenever possible related statutes and regulations (e.g., pharmacy practice act and pharmacy board regulations) were described together to reduce redundancy. All policy provisions identified for this analysis were current as of 30 September 2018.

## 3. Results

### 3.1. Objective 1—Service Process and Implementation

#### 3.1.1. Pharmacy Infrastructure

Prior to offering naltrexone injections, the pharmacies had to create the service infrastructure. Infrastructure development comprised: (a) completing training on how to provide injectable medications in general; (b) working with Alkermes (manufacturer of injectable naltrexone) or other nurse educators to complete naltrexone injection training; (c) obtaining a Clinical Laboratory Improvement Amendment waiver to administer the rapid urine drug screen; and (d) establishing policies and procedures (e.g., urine drug screen testing, naltrexone procurement). Pharmacists also discussed setting aside a private consultation room or area to provide the injections that was located away from the prescription counter and other pharmacy and customer activities.

#### 3.1.2. Summary of How Patients are Referred to the Pharmacist

All of the pharmacists providing the service reported that they had established collaborative practice agreements with a behavioral health practitioner or physician prescriber in accordance with Wisconsin statutes and regulations outlined in Table 1; although prescribing practitioners can refer patients to a pharmacy for naltrexone injections without a collaborative practice agreement, using such a document better assures coordination of care, especially for the provision of long-term care. One pharmacist said that 80% of their patients receiving the injections were from provider referrals, but that 20% of patients were walk-ins. These unscheduled patient walk-ins require immediate pharmacy staff attention creating a coordination burden for the pharmacist and patient, which may conflict with their ongoing counseling and dispensing responsibilities. The pharmacist planning the service said that a referral process from primary care providers whose clinics could not provide the injections would need to be established to bring patients with OUD to the pharmacy for the injections.

#### 3.1.3. Development of a “Straw Model”

Results of the analysis of the data collected from the semi-structured pharmacist interviews were used to develop a “straw-model” (see Figure 1) of the pharmacist-provided naltrexone injection service. The term “straw-model” provides an initial representation of a process that is then utilized to generate discussion and revision [49,50]. The “straw-model” we developed for the naltrexone injection service starts with access and acceptability barriers that may prevent a patient with OUD from using, or their practitioner from referring the patient to, a pharmacist for the naltrexone injection service. The next three steps of the “straw-model” focus on the activities that occur prior to (e.g., scheduling), during (e.g., urine drug screen, injection) and after the actual service encounter (e.g., scheduling follow-up appointment) with the pharmacist.

Access and Acceptability Barriers. The interviews with pharmacists providing the naltrexone injection service identified perceived access and acceptability barriers (Table 3) from both the patient and the pharmacist perspectives. Pharmacists shared comments they heard from patients about barriers to medication and treatment for OUD in general and to naltrexone injections. Access barriers, broadly defined, related to the infrastructure and access issues. Acceptability barriers included issues related to prescribers’ perceptions about treating OUD, and patients’ perceptions about OUD. There was consensus that pharmacists thought patients feel less stigma receiving OUD treatment in the pharmacy since other customers remain unaware of the reasons for the patient presence in the pharmacy.

Injection Service Process Steps. During the interviews, pharmacists outlined the sequence of steps involved in the injection service and provided details about each step, in response to the interviewer’s prompts. Prior to providing a naltrexone injection, the pharmacist must work with the prescriber and/or patient to schedule the appointment, and obtain the naltrexone injection after receiving the order from the prescriber. In some circumstances the drug can be ordered directly from the drug wholesaler. If the naltrexone injection is covered by a private prescription drug insurance plan that places the drug in a specialty drug tier, the pharmacist needs to make arrangements to obtain the drug from the specialty pharmacy.

Pharmacists described a general process involved in providing the injections for a patient that is seeing a psychiatry provider via tele-psychiatry at the pharmacy. Once the patient arrives at the pharmacy, to initiate the visit, the provider sends a code to the pharmacist to start the tele-psychiatry service session that includes just the patient and the provider using an i-pad in a private room in the pharmacy. At the end of the session, the psychiatry provider asks the patient if the injection can be provided to the patient today while the patient is in the pharmacy. If the patient responds affirmatively, the psychiatry provider asks the patient to go and get the pharmacist. Once the pharmacist is in the room, they engage in a three-way conversation between the pharmacist, provider and the patient to tailor the dose for the patient and also communicate with the provider/pharmacist about the dosing. After approval is given for the injection, the pharmacist collects relevant patient demographic information (e.g., age, gender) and conducts a rapid urine drug screen. The purpose of the drug screen is to ensure that a patient is opioid free. After preparing the naltrexone injection, the pharmacist administers the injection and monitors the patient (approximately 30 min) for any adverse reactions to the medication. Typically, the patient remains in the private area of the pharmacy until the monitoring is completed. During their visit to the pharmacy, the patient may receive behavioral health counseling from a counselor prior to receiving the injection or during the post-injection observation period via telemedicine.

After the appointment is completed, the pharmacist will schedule a follow-up appointment with the patient, bill for the service, and communicate the administration of the drug and monitoring feedback to the prescriber. Billing for services for Medicaid patients typically occurred under Healthcare Common Procedure Coding System (HCPCS) code Q-3014 (Telehealth originating site facility fee). One pharmacist indicated that they could charge the patient an injection administration fee if the patient provided the naltrexone medication. Under this scenario, the patient picked up the medication from a specialty pharmacy as required by their private insurer and then transported the medication to the local pharmacy who administered the medication and the patient then paid the administration injection fee directly to the community pharmacy.

#### 3.1.4. Perceived Facilitators and Barriers

Pharmacists identified facilitators as well as internal and external barriers to providing naltrexone injections in the community pharmacy (Table 4). Internal barriers were categorized as those barriers that related to aspects of the pharmacy or pharmacists that were providing the injection. External barriers were categorized as those that occurred outside of the pharmacy and related to patient, community, and health system factors. Since community pharmacies are more convenient for the patient and reduce patient perceptions about stigma, pharmacists in one pharmacy expressed a belief that physicians actually prefer that pharmacies provide the naltrexone injections because it facilitates access to care. However, these same pharmacists believe that insurance drives the locations where naltrexone can be administered, thus limiting patient access to the injections. Internal barriers reflect concerns expressed by the pharmacist about the viability of providing naltrexone injections (e.g., fixed cost investment, liability risks and inadequate reimbursement); and inexperience in providing naltrexone injections or process concerns (e.g., time to coordinate activities, lack of experience and training on how to schedule patients and manage no-shows).

The external barriers generally differ from the access and acceptability barriers described previously in Table 4. Identified external barriers focus on issues such as the absence of resources in the community to support care coordination or to provide wrap-around services that are needed by the patient to adequately manage their chronic disease related to OUD. For example, one pharmacist mentioned a desire to connect patients with a recovery coach in the community to support the patient and improve the likelihood that the patient will return for subsequent naltrexone injections. The final set of external barriers focus on provider and patient misperceptions about their community pharmacy and the available services. For example, residents in the community may believe that the local community pharmacy primarily fills prescriptions in a retail capacity and does not offer clinical services, including naltrexone injections for persons with an OUD.

### 3.2. Objective 2—Analysis of Wisconsin Statutes and Regulations

The content of identified provisions and their implications, as well as the accompanying legal citations, are detailed in Table 5 for each state agency that has relevant activities involving drug abuse treatment and control.

A systematic analysis of Wisconsin statutes and regulations identified a variety of provisions that could facilitate a pharmacist service for providing naltrexone injections to treat patients with OUD. Requirements contained in Correction laws and in Health Services regulations provide ample chances, under a variety of situations, for pharmacists to be a member of the patient care team for OUD. This role is further strengthened by DHS’s authority to maintain treatment coordination for people with drug abuse problems (Wis. Stat. § 51.45). Treatment continuity would be especially important for people transitioning within the corrections, health services, and social services systems, or for inmates who are released into the community. Despite these opportunities, there are instances of legal language that could impede pharmacists providing the injection service. For example, healthcare assistance to needy veterans does not seem to involve pharmacists, even though drug abuse issues can be a problem with which needy veterans are struggling. In addition, telemedicine authorization is not contained in pharmacy practice laws, although it is permitted for physicians. As a result, it is ambiguous whether a pharmacist could be involved in distance consultations with a physician even as a function of a collaborative practice agreement between the two healthcare professionals.

## 4. Discussion

The results of this study suggest that a pharmacist-provided naltrexone injection service can be an access point for the treatment of patients with OUD. Additionally, the legislative and regulatory analysis documented numerous opportunities for the service to be incorporated into the current infrastructure for public health and social services in Wisconsin. Given the shortage of behavioral health providers in Wisconsin, and that pharmacies are more accessible to patients, a pharmacist-provided naltrexone injection service can broaden patients’ treatment options. A pharmacist-provided naltrexone injection service has the potential to be a “game changer” for OUD treatment [51]. The study identified five key factors that explain why MOUD from pharmacists is not widely available: transportation; awareness and acceptance; inter-organizational coordination of care; reimbursement and funding; and service infrastructure including telemedicine.

Transportation to and from the pharmacy for patients needing MOUD is a critical issue that needs to be addressed. Community pharmacists have a long history of offering delivery service to patients to facilitate prescription drug access. Perhaps pharmacists can use their experience with delivery service to develop a cost-effective method to convey patients to and from the pharmacy for naltrexone injections. Another model that could be considered is pharmacists making monthly trips to treatment facilities and providing the injections to groups of patients. Further research is required to understand the needs of patients with OUD to access the injection service and to study the costs and effectiveness of different methods to access the service.

Similar to other pharmacist services, increasing awareness about the service and promoting acceptability are important for the spread of the service. According to the interview results, pharmacists said that patients were universally accepting of the service since it reduced the stigma of treating OUD. However, additional research needs to determine the generalizability of these interview findings, and more broadly assess the perceptions and attitudes of patients with OUD about receiving naltrexone injections from a pharmacist. Learning which aspects of the service—before, during, and after the injection—are most beneficial to patients could help pharmacists to better promote the service and generate better adherence outcomes.

Increasing prescriber acceptance of the service is integral to more widespread adoption and sustainability of the service. A strategy to promote practitioners’ acceptance of the service is informed by the process illustrated by the interviewed pharmacists to provide the naltrexone injections, as well as described in a previous study [52]: (1) establishing and following a protocol that is used in clinics when treating patients with OUD and (2) meeting quality benchmarks to show that the process and service is of high quality. Approaches to promote the acceptability of the service to prescribers could focus on advantages of naltrexone (i.e., it is not a controlled substance, and prescribers do not have to be DATA-waived) and how pharmacists can facilitate access to naltrexone and can provide the injection and post-injection monitoring. As also suggested by our analysis of legislative and regulatory language, public health and social services administrators and affiliated prescribing practitioners need to be accepting of the service as well. Importantly, understanding practitioners’ attitudes and perceptions about pharmacists’ involvement in OUD treatment, including providing naltrexone injections, would be a useful gauge for the viability of the service. Attitudes and perceptions could be used to develop approaches and messaging to increase acceptability of the service. Additionally, researchers and pharmacists are encouraged to propose comparative effectiveness trials of pharmacist-provided naltrexone injections to study the relative advantages of the service. Dissemination of the results could be used to promote acceptance of the service.

One implication of the injection service process is that it requires extensive engagement, communication, and coordination to get the patient into the pharmacy for the first injection, as well as additional coordination to assure their return for follow-up. The finding that coordination and communication is a key component of successful OUD treatment is consistent with research about OUD treatment in clinics [53]. The pharmacists we interviewed had spent significant time working with the behavioral health provider to be part of the patient care team, including mechanisms for communication. Although telemedicine with a behavioral health provider was a component of the injection service described by the interviewed pharmacists, OUD treatment with naltrexone can be initiated by local primary care providers. Pharmacists interested in starting a naltrexone injection service should be aware of OUD treatment providers and access issues in their immediate and surrounding communities. Such efforts should focus on the connection with community resources to promote awareness and acceptance of community pharmacy provided naltrexone injections by non-prescribers.

Interviewed pharmacists reported that reimbursement for costs associated with the administration of the naltrexone injection is not widely available. One pharmacist even provided an itemized list of naltrexone administration costs, and it was estimated that each administered injection resulted in approximately $102 of unreimbursed cost. It should not be surprising that the non-remunerated cost of providing the service can be a substantial barrier impacting the initial decision to provide the service and for the sustainability of the service in a community. Although the analysis of state laws did not involve reimbursement laws due to their complexity, it did identify conceivable opportunities for payment for the injection administration service through separate funding from the public health and social services areas.

As mentioned, there are many additional sources of state-level funding that possibly could help expand pharmacists’ role in OUD treatment across the state, and pharmacists may put themselves in a beneficial position by learning whether possibilities for payment for the injection administration, or other mechanisms for payment, are available through these sources. In fact, an entire statute (Charitable, Curative, Reformatory and Penal Institutions and Agencies, Chapter 46) is devoted to preventing substance abuse, and providing community-based services for people experiencing difficulties with substance abuse issues, primarily through the establishment of program funding opportunities. Table 5 also identifies additional funding opportunities available through DHS-distributed grants, specific drug abuse treatment funds, and veterans’ service grants. Each of these potential funding sources presumably creates the prospect of expanding MOUD around the state. As pharmacist-provided naltrexone injections become increasingly normalized as a convenient and reliable avenue for OUD treatment within the community, it is likely that such funding would broaden the availability of that essential service.

In addition to legally-sanctioned funding opportunities, it is clear from the analysis of statutes and regulations that Wisconsin law provides broad legal capacity, through a variety of government agencies, for the treatment of people with an OUD—even though demand often outstrips available resources. Infrastructures for OUD treatment seem especially robust for the areas of corrections, health service, and social services, as well as through the State Alcohol, Drug Abuse, Developmental Disabilities and Mental Health Act. Although considerable systems exist by law for people needing AODA treatment, the role of pharmacists and their ability to administer injectable naltrexone is currently either undefined or underutilized. The extensive legal foundation for comprehensive AODA assessment and treatment in Wisconsin is still advantageous, because, as this pharmacy service develops and spreads, statutes and regulations could more clearly define the role of pharmacists and facilitate demand for the service.

While Wisconsin law permits pharmacists to give naltrexone injections either pursuant to or without a collaborative practice agreement with a physician, little additional guidance is provided to pharmacists about engaging in such treatment. However, it would be possible for the pharmacy board to develop practice standards for pharmacist-involved OUD treatment, including with injectable naltrexone. Such a standard, potentially coupled with an appointed committee’s advisory statement about OUD-related behavioral health issues, would offer important clarification about the extent of pharmacists’ potential contributions to treatment with naltrexone. In addition, the State Alcohol, Drug Abuse, Developmental Disabilities and Mental Health Act identifies “methadone maintenance programs” as including the use of naltrexone; although there are no legislative notes to provide information about the contributions that pharmacy service can make to these programs, agency policies and procedures could be modified to specifically describe the role of pharmacists providing treatment in methadone maintenance programs. As such results suggest, clarifying these standards could contribute to promoting awareness of the pharmacist service, as well as better assure that health and social services administrators and practitioners are more accepting of the service.

In relation to general healthcare practice, physicians can engage in telemedicine to facilitate distance treatment. When pharmacists and physicians collaborate to conduct a visit via telemedicine, as done by the pharmacists we interviewed, it offers a new location (i.e., community pharmacy) for the provider-patient visit. The process then facilitates the pharmacist providing the naltrexone injection. The lack of a parallel telemedicine provision in pharmacy practice laws, absent clarifying authoritative statements, creates uncertainty about the legality of pharmacists’ involvement in physician or patient interactions through telemedicine consults. To establish well-defined authority for pharmacists to engage in telehealth, the pharmacy examining board could modify its regulations to permit such practice. Such regulatory change, coupled with a communications strategy to its licensees about the change, has the potential to facilitate pharmacist service around naltrexone treatment for patients with OUD and expand access to MOUD treatment.

### Limitations

A few limitations characterize this initial analysis of state laws. First, it is possible that relevant provisions were contained in either statutes or regulations but were overlooked or inappropriately disregarded during the review. Second, due to the complexity of reimbursement-related laws (at the Federal, state, and local levels, and either public or private sector), description of state insurance laws were excluded for this project. Since the applicability of state insurance laws does not function in isolation of other types of reimbursement laws, a more detailed discussion of these policies is necessary for providing an accurate account of the coverage of pharmacy-provided naltrexone treatment, as well as providing telemedicine services, which was outside the scope of this article. However, further research is indeed necessary to examine the potential influence of the variety of reimbursement-related laws affecting the use of injection naltrexone for OUD treatment.

Our understanding of the process of community pharmacists providing naltrexone injections was based on a purposeful sample, and thus has a few limitations. First, it is possible that there are other community pharmacists in Wisconsin who provide naltrexone injections and use service processes that vary from the initial straw model we developed. Research to further describe the service process is needed. Second, the perceived prescriber and patient acceptability barriers, as well as the identified facilitators and barriers, may be understated, which in turn could impact the willingness to utilize community pharmacists as a provider of naltrexone injections. Further research is needed to conduct a broad environmental scan of community pharmacists, including if they are currently providing (or would be willing to provide) naltrexone injections and identifying other perceived facilitators and barriers to offering the service. Also, additional information is needed about the existing infrastructure in community pharmacies offering naltrexone injections. The information could be used to develop a toolkit for other community pharmacists and the toolkit’s effectiveness could be examined in a comprehensive dissemination and implementation research study design.

## 5. Conclusions

The nascent growth of the partnership between community pharmacists, individuals with an OUD and prescribers in Wisconsin to offer naltrexone injections highlights the potential for community pharmacists to be active partners in addressing issues related to the growing opioid crisis. However, barriers associated with transportation, service infrastructure, reimbursement, awareness and acceptance by practitioners including communication and service coordination need to be studied to facilitate implementation and sustainability of this service in community pharmacies. In addition, the design, implementation and effectiveness of current naltrexone injection service delivery approaches by community pharmacists are not well understood. Operating in a supportive legislative and regulatory environment, community pharmacists, as an already trusted and acceptable provider of service within their community, could provide individuals a significant access point for OUD treatment with naltrexone injections.

## Figures and Tables

**Figure 1 pharmacy-07-00059-f001:**
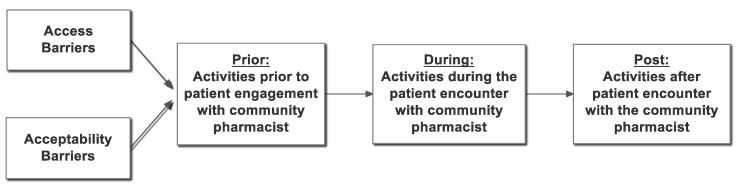
Community Pharmacy Medication Assisted Treatment with Naltrexone “Straw-Model”.

**Table 1 pharmacy-07-00059-t001:** Wisconsin State Statutes and Regulations Related to Pharmacist Authority to Provide Patient Care Services.

Wisconsin State Law Section	Intended Purpose of Cited Section of Wisconsin State Law
Wisconsin statute for Medical Practices [Section 448.03(2)]	Authorizes physicians to delegate patient care services to other health care providers through a collaborative practice agreement between physicians and pharmacists.
Wisconsin statute for Pharmacy Practice [(Section 450.033)]	Provides statutory authority for pharmacists to perform any patient care service delegated to a pharmacist by a physician
Wisconsin statute for Pharmacy Practice [(Section 450.035 (1r)]	Provides statutory authority for pharmacists to administer ***non-vaccine drugs*** via injection after completing specific training
Wisconsin statute for Pharmacy Practice [(Section 450.035 (1t))]	Provides statutory authority for pharmacist interns to administer ***non-vaccine drugs*** via injection after completing specific training
Wisconsin Pharmacy Examining Board (in Chap Phar 7 Section 7.10)	Establishes additional requirements for pharmacists and pharmacist interns who are administering ***non-vaccine drugs*** via injection

**Table 2 pharmacy-07-00059-t002:** Reviewed Statutes and Regulations that can Influence Pharmacist-Provided Medication-Assisted Treatment with Naltrexone.

Legislation	Regulations
Veterans Chapter 45	Department of Veterans Affairs
Social Services Chapter 46	
State Alcohol Drug Abuse, Developmental Disabilities, and Mental Health Act Chapter 51	
Chapter 150 Regulation of Health Services, Subchapter IV	Department of Health Services
Corrections Chapter 301	Department of Corrections
Regulation and Licensing Chapter 440 for prescribers Chapter 450 for pharmacists	Department of Regulation and Licensing
	Department of Children and Families

**Table 3 pharmacy-07-00059-t003:** Access and Acceptability Barriers to Pharmacist-provided Naltrexone Injection Service.

Access Barriers	Acceptability Barriers
Infrastructure - Lack of clinic infrastructure to support providing injections - Prescriber unaware of naltrexone - Naltrexone covered as a Specialty Medication—limited pharmacy access	Prescriber Perceptions about Treating OUD - Patients with OUD are difficult to manage - Prescriber concerns about the use and potential diversion of medications
Access - Lack of transportation to prescriber and/or pharmacy (especially in rural areas) - Poor Access to prescribers (especially rural areas) - Access to naltrexone Injections	Patient perceptions - Stigma associated with MAT - Patient knowledge and fear about medication

**Table 4 pharmacy-07-00059-t004:** Naltrexone Administration: Preliminary Facilitators and Barriers in Community Pharmacies.

Facilitators	Internal Barriers	External Barriers
Motivated pharmacists and pharmacy ownership structure	Frontend fixed costs associated with staff, training, remodeling, on-site drug testing and billing.	Lack of adequate patient transportation and care coordination with prescriber, caseworker and other entities leads to poor adherence
Patient trust in the community pharmacist	Lack of a business case including insufficient reimbursement for drug administration and testing	Lack of supportive wraparound services (e.g., behavioral health)
Pharmacists’ and behavioral health providers’ knowledge of telemedicine and its role in proving health care.	Time to coordinate activities associated with administration including prior authorization, patient scheduling & managing appointment no-shows	Lack of awareness on ability to refer patients to pharmacies via collaborative practice agreement.
Flexible scheduling: community pharmacy vs. physician office to provide injections	Liability risks associated with providing MAT in the pharmacy	Misperceptions: pharmacists do not provide patient services/not treatment team member
Availability of training courses that allow pharmacist to meet regulations regarding injections	Pharmacists’ lack of training and experience in MAT injections and induction.	Patient reluctance to pay drug co-pays or pharmacy injection fees
Willingness of pharmacists already engaged in practice to share knowledge with others		Pharmacy seen as a retail establishment versus clinic service provider

**Table 5 pharmacy-07-00059-t005:** Statutory and Regulatory Provisions Relevant to Pharmacist Services for Naltrexone Injections.

Section of Law (Statutory Citation) (Regulatory Citation)	Description of Identified Provisions	Implication for Pharmacist Service
Children and Families (Statutes—none) (Regulations—Wisconsin Administrative Code; Department of Children and Families)	Applicants for work experience programs require substance abuse screening, testing, and referrals treatment (Wis. Adm. Code DCF 105.01), and a positive test requires treatment participation (Wis. Adm. Code DCF 105.06)	Requirements could increase demand for naltrexone injection service
Corrections (Statutes—Corrections) (Regulations—Wisconsin Administrative Code; Department of Corrections)	Prisoners are provided drug abuse assessment and treatment at each facility within the corrections system, while parolees or people on extended supervision also are to receive drug testing (Wis. Stat. § 301.03). For healthcare services, a prescription drug formulary is used (Wis. Stat. § 301.103), but covered medications are not specified	If naltrexone is included on the drug formulary, it conceivably could be offered through pharmacist service as a viable modality if positive assessments lead to treatment
	Substance abuse treatment also can seem to be a component of a variety of correctional programs and services, including: for inmates selected for the challenge incarceration program (Wis. Stat. § 302.045; Wis. Adm. Code DOC 302.38(3)(e)) for inmates transferred from state prisons (Wis. Stat. § 302.05) those in an earned release program (Wis. Adm. Code DOC 302.39(3)(c)) for those on-site or in hospitals using crisis intervention services (Wis. Stat. § 302.365) when transferring a prisoner to a hospital or an approved treatment facility (Wis. Stat. § 302.38) within residential, outpatient, and aftercare settings, as a means to reduce recidivism (Wis. Stat. § 301.068) when in intensive sanctions program, as an alternative to incarceration (Wis. Adm. Code DOC 333.01(3) & (4)) when transferring to the Wisconsin resource center (a specialized treatment program for inmates in need of mental health services) (Wis. Stat. § 302.055) for those in jail using a prisoner classification system to provide services and programs based on medical/mental health needs (Wis. Stat. § 302.36; Wis. Adm. Code DOC 302.02) for those in municipal lockup facilities and jails using a health screening form to identify drug abuse problems (Wis. Adm. Code DOC 349.03(9) & (17); Wis. Adm. Code DOC 350.03(12) and policies and procedures to provide drug abuse treatment services (Wis. Adm. Code DOC 349.16(1)(b); Wis. Adm. Code DOC 350.13(1) & (2))	There are a variety of opportunities throughout the corrections system for identifying inmates as needing substance abuse treatment, and to potentially engage pharmacists to provide naltrexone injections either directly through the corrections facility or through their community pharmacies
	Under certain circumstances, the DOC must notify local law enforcement before releasing a person into extended supervision (Wis. Stat. § 302.113) and can facilitate inmate release (Wis. Adm. Code DOC 302.34(5)(e) & (7)(g); Wis. Adm. Code DOC 302.35(3)(e)(2)) and can even expedite a risk reduction sentence (Wis. Adm. Code DOC 302.40(3)(e))	Such notification offers an opportunity to coordinate substance abuse treatment needs within the community, which could involve pharmacist naltrexone injection services if pharmacists are identified as a viable community resource
Health Services (Statute—none) (Regulation—Wisconsin Administrative Code; Department of Health Services)	Wisconsin counties must provide emergency mental health services (Wis. Adm. Code DHS 34.01), for conditions contained in the American Psychiatric Association’s Diagnostic and Statistical Manual (Wis. Adm. Code DHS 34.02(14))	The American Psychiatric Association’s Diagnostic and Statistical Manual includes an OUD diagnosis, providing a clear context for pharmacists’ naltrexone injection service
	A number of specific health services programs and services permit SUD treatment, including: outpatient mental health clinic services (Wis. Adm. Code DHS 35.17(1)(b)(4)) comprehensive community services programs (Wis. Adm. Code DHS 36.02) treatment alternative program (TAP) for people involved in the criminal justice system, as a means to avoid imprisonment (Wis. Adm. Code DHS 66.01(1)) for applicants of certain employment and training programs (Wis. Adm. Code DHS 38.06(1) & (2)) for people with an SUD diagnosis who have a functional impairment that interferes with major life activities (Wis. Adm. Code DHS 36.14(1) & (2)) community support programs (Wis. Adm. Code DHS 63.08(1) community substance abuse prevention and treatment services (Wis. Adm. Code DHS 75.01(1)) prevention services and strategies to reduce the risk of substance abuse (Wis. Adm. Code DHS 75.04) emergency outpatient service (Wis. Adm. Code DHS 75.05(1)) medically-managed inpatient treatment service (Wis. Adm. Code DHS 75.10(1)) individual and group counseling (Wis. Adm. Code DHS 75.10(6)(f)) day treatment service (Wis. Adm. Code DHS 75.12(1)) outpatient treatment service (Wis. Adm. Code DHS 75.13(1)) transitional residential treatment service (Wis. Adm. Code DHS 75.14(1) & (6)(b))	There are a variety of opportunities throughout the Health Services system for identifying people as needing substance abuse treatment, all of which could be used to coordinate naltrexone treatment with community pharmacists
	opioid treatment service for OUD providing methadone or other FDA-approved medications, as well as other medical or psychological services, counseling, or social services (Wis. Adm. Code DHS 75.15(1)) intervention services that includes case management (Wis. Adm. Code DHS 75.16(1))	This provision provides a direct role for pharmacists and their authorization to provide naltrexone injections for the treatment of OUD
Regulation and Licensing (Statutes—Regulation and Licensing; Chapter 448. Medical Practices/Chapter 450. Pharmacy Examining Board) (Regulations—Wisconsin Administrative Code; Medical Examining Board/Pharmacy Examining Board)	Healthcare examining boards can establish practice standards (Wis. Stat. § 450.02)	Practice standards could include pharmacist services in providing naltrexone injections
	Advisory committees can be convened to address behavioral health issues (Wis. Stat. § 440.043)	An advisory committee could be convened to address pharmacist services for OUD prevention and treatment
	Any licensed physician can use telemedicine as a patient engagement tool, after documenting a patient evaluation (Wis. Adm. Code Med 24.07)	Telemedicine authorization does not involve pharmacists, and it is unclear how this provision extends to pharmacists who are part of a collaborative agreement with a physician
Social Services (Statutes—Charitable, Curative, Reformatory and Penal Institutions and Agencies; Chapter 46. Social Services) (Regulations—none)	DHS has established a drug abuse program that creates the foundation for education, diagnosis, and treatment (Wis. Stat. § 46.973), and county-level DHS offices are developed to address, in part, drug abuse issues (Wis. Stat. § 46.23) Through a variety of funding mechanisms, community-based drug abuse prevention and treatment can focus on residential care, prisoner reintegration into communities, urban communities, and underserved populations (Wis. Stat. § 46.48), as well as to facilitate long-term care transitions (Wis. Stat. § 46.2803), for low-income Hispanics and Black Americans in urban areas, the Native American population, and women (Wis. Stat. § 46.975), and for inmates in the criminal justice system as an alternative to imprisonment (Wis. Stat. § 46.65)	Implementation of community-based program funding could increase demand for naltrexone injection service, especially when DHS efforts acknowledge the role of and establish relationships with community pharmacists that provide those services
State Alcohol, Drug Abuse, Developmental Disabilities and Mental Health Act (Statutes—Charitable, Curative, Reformatory and Penal Institutions and Agencies; Chapter 51. State Alcohol, Drug Abuse, Developmental Disabilities and Mental Health Act) (Regulations—none)	This statute addresses a broad range of AODA prevention and treatment services and is designed to assure continuity of care for such treatment (Wis. Stat. § 51.001), which reinforces DHS’s authority to establish a comprehensive and coordinated drug abuse program for education, diagnosis, and treatment (Wis. Stat. § 51.45)	Pharmacist-provided drug abuse treatment services could be a regular component of DHS coordinated care efforts
	Methadone treatment programs include the provision of methadone, buprenorphine, and naltrexone (Wis. Stat. § 51.4223)	The provisions for methadone treatment programs could allow pharmacist services for naltrexone injections
Veteran’s Affairs (Cultural and Memorial Institutions; Veteran’s Affairs; Chapter 45. Veterans) (Wisconsin Administrative Code; Department of Veteran’s Affairs)	Healthcare assistance from a variety of health care providers is available to all needy veterans (Wis. Stat. § 45.40)	Under this section, the definition of “health care provider” does not include pharmacists
	Substance abuse treatment programs approved by the U.S. Department of Veteran’s Affairs (USDVA) or Wisconsin-certified AODA programs are available for needy veterans (Wis. Adm. Code VA 2.01), and treatment in such programs can facilitate subsistence aid when veterans lose income due to drug abuse (Wis. Adm. Code VA 2.01(3)(b))	Given the description of AODA-related programs in Wisconsin’s Health Services and Social Services regulations, as well as in the State Alcohol, Drug Abuse, Developmental Disabilities and Mental Health Act, it is likely that such treatment could involve injection naltrexone
	Federal grant to counties can be issued to improve services to veterans (Wis. Stat. § 45.82), and the Tribal veterans’ service office can apply for American Indian grants (Wis. Adm. Code VA 15.02(1))	Veteran-related funding may potentially be applied to drug abuse issues, but pharmacist involvement in providing such services may be limited due to their not being a recognized “health care provider”

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
