# Peer review of "Systematic Analysis of the Service Process and the Legislative and Regulatory Environment for a Pharmacist-Provided Naltrexone Injection Service in Wisconsin"

_pharmacy, 2019, doi:10.3390/pharmacy7020059_

Round 1

Reviewer 1 Report

This is an interesting qualitative study of MOUD in Wisconsin community pharmacies. Several comments will strengthen the manuscript. At best, it should be described as a pilot study.

line 67 - briefly describe DATA-waived physicians.

line 135 - 142 - for sample selection, was it possible to query the Wisconsin Board of Pharmacy for those pharmacists with injection license? Were all 9 pharmacists providing MOUD contacted, and if so, why did 5 not participate? The sample needs to be described as purposeful not convenience. There is no mention of IRB review.

line 144 - suggest 'community' as the descriptor of pharmacy in the ambulatory setting, not retail

line 144 - suggest changing  'We conducted a total of three interviews with pharmacists' to 'We conducted one interview with three pharmacists'

lines 144-151 - it's not clear how many pharmacists were involved in the study. Needs rewritten.

lines 152-157 - most qualitative study interviews are recorded, transcribed verbatim, and analyzed systematically with a qualitative software package, like Atlas.ti or NVivo. Including actual statements from pharmacists as examples would strengthen the rigor of the manuscript.

lines 193-195 - omit these sentences, as the breakdown follows from the objectives stated in the methods section.

lines 212-213 - how did they describe connecting walk-ins with a practitioner?

lines 247- 271 - is this the author's summary of one pharmacist's process? Is this consistent among the others?

Table 5 is an excellent summary and analysis of legal authority for pharmacists to provide MOUD.

lines 321 - 480 - the discussion, while interesting, is way too long, and additional data collection from rural public health personnel was not described in the methods..The first paragraph could outline the key factors that explain why MOUD from pharmacists is not widely available. That is, transportation and telemedicine for MOUD, awareness, reimbursement, standards-setting, and provider status.

lines 482 - 489 - the conclusion should follow from the findings. Why don't pharmacists provide MOUD in Wisconsin, given the supportive legal environment? The authors need to state what they found, and not what is potential.

References are not in mdpi format.

Author Response

Response to Reviewer 1 Comments

Point 1: This is an interesting qualitative study of MOUD in Wisconsin community pharmacies. Several comments will strengthen the manuscript. At best, it should be described as a pilot study.

Response 1: We added the word “exploratory” in lines 20, 110 and 115 to address this concern.

Point 2: line 67 - briefly describe DATA-waived physicians.

Response 2: We added a brief sentence in lines 67 to 68 to briefly describe the term “DATA-waived physicians”.

Point 3: line 135 - 142 - for sample selection, was it possible to query the Wisconsin Board of Pharmacy for those pharmacists with injection license? Were all 9 pharmacists providing MOUD contacted, and if so, why did 5 not participate? The sample needs to be described as purposeful not convenience. There is no mention of IRB review.

Response 3: We were not able to query the Wisconsin Board of Pharmacy about pharmacist with an injection license because the information was not publically available to our team.

We also restructured the opening of this paragraph in lines 144 to 146 to clarify our sampling approach and remove the sentence about April 2019 and 9 pharmacists which was included in error.

As suggested, we replaced “convenience” with “purposeful” in line 450 and added the qualifier in our revised sampling strategy description.

We discussed this project with the UW-Madison Health Sciences IRB. In preparing the manuscript, we talked with the five pharmacists as stakeholders who provided feedback about a process, the provision of naltrexone injections in a community pharmacy. The intent of these conversations was not to produce generalizable findings and they did not represent a systematic investigation intended to address specific research aims or questions. The federal definition of research is “a systematic investigation, including research development, testing and evaluation, designed to develop or contribute to generalizable knowledge.” Per our conversations with the IRB, they informed us that if the pharmacists are providing feedback about a process and not providing information about themselves or undergoing protocol dictated procedures to create generalizable knowledge, we (the IRB) would not consider them subjects and thus no IRB approval was needed because our project did not meet the federal definition of research. Text was added in lines 155 to 157 to address this issue related to the IRB

Point 4: line 144 - suggest 'community' as the descriptor of pharmacy in the ambulatory setting, not retail

Response 4: The change was made as suggested

Point 5: line 144 - suggest changing  'We conducted a total of three interviews with pharmacists' to 'We conducted one interview with three pharmacists'

Response 5: Sentence was changed and now reads “We conducted four interviews with a purposeful sample of five pharmacists… as part of our restructuring of how we described the sampling strategy.

Point 6: lines 144-151 - it's not clear how many pharmacists were involved in the study. Needs rewritten.

Response 6: We restructured the opening of this paragraph in lines 144 to 146 to clarify our sampling approach

Point 7: lines 152-157 - most qualitative study interviews are recorded, transcribed verbatim, and analyzed systematically with a qualitative software package, like Atlas.ti or NVivo. Including actual statements from pharmacists as examples would strengthen the rigor of the manuscript.

Response 7: We restructured the paragraph related to data analysis (lines 154 to 163) to reflect our approach given that the sampling approach involved a purposeful sample of community pharmacists who we engaged in a conversation to learn about their experience, as experts, in providing naltrexone injections

Point 8: lines 193-195 - omit these sentences, as the breakdown follows from the objectives stated in the methods section.

Response 8: Sentences were deleted as suggested.

Point 9: lines 212-213 - how did they describe connecting walk-ins with a practitioner?

Response 9: We appreciate the opportunity to provide clarification relates to walk-ins. The issue was more associated with the burden that unscheduled patients with a valid naltrexone prescription place on the current processes within the community pharmacy. We restructured the text in lines 215 to 220 to address this issue.

Point 10: lines 247- 271 - is this the author's summary of one pharmacist's process? Is this consistent among the others?

Response 10: We changed the introduction to this paragraph in lines 250 to 251 to provide information that what follows is a general description of the injection process.

Point 11: Table 5 is an excellent summary and analysis of legal authority for pharmacists to provide MOUD.

Response 11: Feedback received during the review process about the value of Table 5 versus its length differed between reviewers. The information in the Table summarized relevant statutory and regulatory provisions related to pharmacy services for naltrexone injections. To address both concerns, we carefully reviewed the table to remove any unnecessary words and reformatted the table in an effort to shorten its overall length from 4.5 pages to 4 pages in the manuscript.

Point 12: lines 321 - 480 - the discussion, while interesting, is way too long, and additional data collection from rural public health personnel was not described in the methods. The first paragraph could outline the key factors that explain why MOUD from pharmacists is not widely available. That is, transportation and telemedicine for MOUD, awareness, reimbursement, standards-setting, and provider status.

Response 12: We appreciated the feedback regarding the overall length of the Discussion section. As suggested we restructured the introductory paragraph (lines 324 to 333) to indicate the key factors that would be addressed in the Discussion section. We removed the sentence referring to rural public health personnel. We also carefully reviewed the Discussion and removed non-essential material resulting in a reduction from 283-word reduction (1786 to 1503).

Point 13: lines 482 - 489 - the conclusion should follow from the findings. Why don't pharmacists provide MOUD in Wisconsin, given the supportive legal environment? The authors need to state what they found, and not what is potential.

Response 13: The opportunity for community pharmacists to become active partners in addressing the opioid crisis in Wisconsin is an important conclusion from this research. We restructured the conclusions to reflect the importance of our findings in response to the reviewer feedback.

Point 14: References are not in mdpi format.

Response 14: We updated the references to mdpi format via endnote

Reviewer 2 Report

This is an excellent paper, providing a very good analysis on a new pharmacy service, critical to deal with a public health emergency. It reads through wonderfully.

-the lenght of table 5. Maybe you could find a way to make it shorter.

There is something that can be added (a question of small details)

- in the objectives section, it could be stated that this is a exploratory study in a clearer way. One understands it is, but maybe some readers would benefit to see it clearly written.

- line 141 states that there are 9 pharmacists/pharmacies identified in April 2019. But in line 146 it says that "interviews took place between April and June 2018". How may pharmacists were there in 2018? Did the numbers change meanwhile? How many were there per pharmacy?

- Were the interviews audio recorded? Did the authors considered to prepare a focus group instead of person-to-person interview? 

- In the conclusion, a clearer statement about the service process, barriers and facilitators found, could be added.

Author Response

Response to Reviewer 2 Comments

Point 1: This is an excellent paper, providing a very good analysis on a new pharmacy service, critical to deal with a public health emergency. It reads through wonderfully.

Response 1: We appreciate the overall positive feedback regarding this manuscript.

Point 2: -the length of table 5. Maybe you could find a way to make it shorter.

Response 2: Feedback received during the review process about the value of Table 5 versus its length differed between reviewers. The information in the Table summarized relevant statutory and regulatory provisions related to pharmacy services for naltrexone injections. To address both concerns, we carefully reviewed the table to remove any unnecessary words and reformatted the table in an effort to shorten its overall length from 4.5 pages to 4 pages in the manuscript.

There is something that can be added (a question of small details)

Point 3: - in the objectives section, it could be stated that this is an exploratory study in a clearer way. One understands it is, but maybe some readers would benefit to see it clearly written.

Response 3: We added the word “exploratory” in lines 20, 110 and 115 to clarify this point for the reader.

Point 4: - line 141 states that there are 9 pharmacists/pharmacies identified in April 2019. But in line 146 it says that "interviews took place between April and June 2018". How may pharmacists were there in 2018? Did the numbers change meanwhile? How many were there per pharmacy?

Response 4: We restructured the opening of this paragraph in lines 144 to 146 to clarify our sampling approach and remove the sentence about April 2019 and 9 pharmacists which was included in error.

Point 5: - Were the interviews audio recorded? Did the authors considered to prepare a focus group instead of person-to-person interview?

Response 5: We added language in line 155 to indicate that the interviews were not recorded. Since the interviews represented a purposeful sample of 5 pharmacists located across a wide geographic area in the state who were currently providing naltrexone injections, conducting a focus group was not considered.

Point 6: - In the conclusion, a clearer statement about the service process, barriers and facilitators found, could be added.

Response 6: We added language in the Conclusion in lines 467 to 469 to address this comment

Round 2

Reviewer 1 Report

The authors have addressed the points for editing. Good job.